# Experimental Study of the Pressures and Points of Application of the Forces Exerted between Aligner and Tooth

**DOI:** 10.3390/nano9071010

**Published:** 2019-07-12

**Authors:** Francesca Cervinara, Claudia Cianci, Francesco De Cillis, Giovanni Pappalettera, Carmine Pappalettere, Giuseppe Siciliani, Luca Lombardo

**Affiliations:** 1School of Orthodontics, University of Ferrara, 44121 Ferrara, Italy; 2Department of Mechanics, Mathematics and Management, Polytechnic University of Bari, 70120 Bari, Italy

**Keywords:** aligner, pressure, force, microcapsules technology

## Abstract

The analysis of forces, moments and pressure points has long been of great interest in orthodontics. Hence, we set out to define a method for measuring the pressure exerted by aligners on the teeth, and specifically to identify the precise points of pressure exertion. Intraoral scans were performed on a patient with optimal alignment and levelling before and after 2º vestibularisation of the upper central incisor. Pressure sensor film was placed in a dedicated housing between the aligner and teeth in order to record the pressure exerted after 15 s of aligner application. The images captured by the film were scanned, digitised, and subsequently analysed. Areas and amounts of pressure generated by the aligners were evaluated, and the net force of each was calculated, adjusted to take into consideration passive values. The method revealed the areas of contact by which the aligner transmits force on the teeth, and the pressures at which it does so. The pressure exerted by an aligner is not evenly distributed across the entire surface of the tooth during lingual tipping of an upper incisor. The areas of force concentration were not identical, as these are influenced by factors resulting from the manufacturing and casting processes.

## 1. Introduction

The analysis of forces, moments and pressure points has long been of great interest in orthodontics. Wolf [1], Moss [2] and Von Limbourg [3] explained how pressure models the bone, directs its growth, and orients the trabeculae, and that anomalous pressure from the lips and cheeks is a determining factor in the genesis of malocclusion. Indeed, the surrounding soft tissues influence tooth position, in particular, the resting equilibrium established by the perioral muscles and tongue. Once this equilibrium is affected by external factors such as bad habits or orthodontic forces, the position of the teeth is altered. 

The literature informs us that an optimal system of forces produces an optimal reaction in the periodontal ligament [4]. However, there are different opinions regarding the nature of the force that is translated into optimal mechanical conditions within the periodontal ligament for orthodontic tooth movement, and the concept of optimal force has changed considerably over the years. For example, Schwarz [5] defined an optimal continuous force which provokes a change in tissue pressure similar to the blood pressure in the capillaries in the periodontal ligament, thereby impeding their occlusion. However, optimal force is more commonly defined as that which generates a maximum rate of orthodontic movement, without damaging the surrounding tissues or causing patient discomfort. This force, however, will vary among both patients and individual teeth [6]. In fact, in 2003, a review by Kuijpers-Jagtman highlighted four fundamental obstacles to defining the concept of optimal force: the difficulty in precisely calculating the distribution of forces in the periodontal ligament with respect to the forces applied; the type of movement and the rate at which this is achieved; the variability in the stress distribution in function of uncontrolled tooth movement; the confusion regarding the relationship between force and rate of tooth movement (considering that orthodontic tooth movement can be divided into several phases (Burstone)) [7]; and the great inter-individual variability in both animals and humans (Von Bohl et al. and Maltha et al.) [7,8].

In the era of aesthetic orthodontics, orthodontic aligners are a popular treatment option for malocclusion. Ideally, aligners are worn for 22 h per day, being removed solely for eating and oral hygiene procedures. They are generally replaced every 7 or 14 days, with the consequent application of new pressures on the teeth. Nowadays, there are various types of aligners, and their characteristics are strongly influenced by their construction material, thickness and fit. Despite this great variability, the aim of this study was to determine the specific pressures that they exert on the teeth, and the precise areas at which these are applied. A secondary objective was to study the reproducibility of the pressures, and therefore forces exerted, by analysing the same aligners in triplicate and taking into consideration manufacture-related variables.

## 2. Materials and Methods

We performed intraoral scans (TRIOS-3Shape) on a patient with molar and canine Class I and optimal alignment and levelling before (T0) and after (T1) 2º vestibularisation of the right upper central incisor (1.1) with a centre of resistance 10 mm from the alveolar crest. Setup was performed using 3Shape OrthoAnalyzer^®^ software (TRIOS 3Shape, Copenhagen, Denmark), and six digital models in STL format were generated, 3 at T0 and 3 at T1 (Figure 1). 

These were in turn used to 3D print low-viscosity resin casts (E-Denstone Material) with a resolution of 35 microns, in a controlled environment at a constant temperature of 140 °C, by means of an EnvisionTEC 3D printer (Desktop XL, PixCera, Perfactory^®^ 4DDP4 Series- Gladbeck, Germany- Dearborn, Michigan, MI, USA).

Before making the aligners, a 180-micron housing for the pressure sensor was created on the surface of the 1.1 and the corresponding gingival area on the casts (Figure 2). This procedure was carried out using a dedicated and extremely precise software (SINERGIA, Nobil-Metal^®^) in order to allow subsequent recordings to be as realistic and faithful as possible. The modified casts were then used as moulds for six aligners, three identical active (A1, A2, A3) for the prescribed movement, and three passive (P1, P2, P3) for calibration purposes. The aligners used in this study were F22 (Sweden and Martina Due Carrarae, Padua, Italy), which are 0.75 mm thick and made of thermoplastic polyurethane (TPU). This material is 20% more elastic than PET-G, and presents a less steep and smaller force decay over 24 h than the same, not to mention an almost two-fold fracture resistance [9]).

In order to measure the pressures transmitted by the aligners to the tooth, we inserted Fuji Prescale Film [10,11,12,13], a chemically treated PET film which is very sensitive to applied pressure, into the housing previously created. The film responds to pressure by turning from milky white to different shades of pink and red; the colour density indicates the amount of pressure applied, thereby providing a kind of pressure map of the tooth surface, a method that is both rapid and repeatable. In fact, in order to correctly optimise this technology for different operational pressures, Fuji Film has defined three measurement fields with specific ranges of action; for the purposes of this study we opted to use the Low Pressure (LP) film (2.5–10 MPa) for the active phase and Super Low Pressure (SLP) film (0.5–2.5 MPa) for the passive phase.

Films were removed 15 s after their application and subsequently analysed. The minimum area that can be measured is 0.1 mm^2^, as the relative distribution of the uniform Fuji Prescale Film microcapsules is 0.1 mm^2^ [10]. However, an accurate analysis of exposed Prescale film was carried out using an FPD-8010E analysis system (Fuji Film, Tokyo, Japan). This method involves scanning the exposed Prescale films, digitising the resulting scans, and storing the images prior to analysis [10,11]. This system is accurate and repeatable, and enabled us to measure both the area and the degree of pressures exerted simultaneously [10]. As in this study we focused on an aligner programmed to exert an active vestibulo-lingual force on the right upper central incisor, we examined the pressure map of the vestibular surface of the same tooth in all cases. Indeed, the upper central incisor has the flattest and most regular vestibular surface, making it suitable for this experiment, which was comprised of two major phases (Figure 3): calibration with three passive aligners (P1, P2 and P3) and analysis of the active phase with three active aligners (A1, A2 and A3). 

Calibration was carried out for all three passive aligners (P1, P2 and P3) on the cast at T0. The Super Low Pressure Film (SLP—0.5 to 2.5 MPa) was positioned in the 180-micron housing in the vestibular surface of the cast, and the aligner was fitted and kept in place for 15 s, before removing the aligner and mapping the film. In total, ten pressure films were obtained for each aligner in order to obtain mean and maximum calibration values to later be subtracted from the subsequently obtained active pressure values subsequently obtained (Table 1). Thus, all measurements were performed ten times, each by a sole operator on three aligners with identical programmed movements on a single cast (cast A at T0). The maximum pressure recorded during the passive phase was 1.46 ± 0.50 MPa. The calibration areas were 1.5 × 1.5 mm^2^ each.

The active analysis was carried out in the same way, ten times for each of the three identical active aligners (A1, A2 and A3) on a single cast (model A in T0) by a sole operator, but this time using a film able to register greater pressures, i.e., Prescale Low Pressure Film (LP—2.5 to 12.75 MPa) (Table 2). 

We then compared the coordinates of the active and passive pressures to ensure that the calibration areas were perfectly superimposed (with reference to Cartesian axes corresponding to the mesial surface and incisal margin of tooth 1.1). Using FPD-8010E software, we then calculated the mean and maximum pressures exerted by the active aligners on these areas with respect to the passive calibration data (Table 3). Not all of the active areas of pressure were associated with areas of pressure exerted during the passive phase.

Hypothesising that the vestibular surface of the tooth examined was flat (i.e., that the tooth was a parallelepiped), and that the forces exerted by the aligner are parallel and perpendicular to the tooth surface, the total net force exerted by each of the three aligners was calculated; areas of pressure measuring 1 × 1 mm^2^ were considered, using the adjusted mean pressure (i.e., the mean active pressure minus the mean passive pressure) calculated for each. The modulus of the resulting force was then calculated using Equation (1), assuming that the force was applied at the centre of the area of pressure.
(1)F=PxA

The point of application of the net force exerted on the entire surface of the tooth was determined by calculating the net force from the first two forces, and adding each of the net forces identified in turn. The net of two parallel forces was found to be applied at a certain point, *P*, which was identified via the following proportion:(2)d1:d2=F1:F2
where, *d*1 and *d*2 indicate, respectively, the distance from point *P* to the points of application *F*1 and *F*2. If the forces are in agreement, point P will be located on the straight line that joins the points of application *F*1 and *F*2, and the net force modulus will be given by the sum of the force moduli for *F*1 and *F*2 as follows:(3)FTot=F1+F2

## 3. Results

As the tables show, there was variability among the aligners studied, despite all efforts to standardise the study protocol. Indeed, the pressure distribution displayed by the three aligners was similar, but not identical, even in the calibration phase. However, the mean net pressure exerted across the entire tooth surface by the active aligners remained constant at roughly 15 MPa. The coordinates of the 5 areas of pressure were also almost identical, and coincide when superimposed (Figure 3). The overall similarity between aligners is confirmed by visual comparison of their respective pressure maps, but the numerical values obtained show that there were differences between the three. In particular, Table 4 reveals the differences between the three aligners in terms of net force and its point of application (with reference to a Cartesian grid whose origin is at the junction of the surface and incisal margin of the tooth). It should be noted that the net force calculated in this way is based on an idealised (i.e., flat) coronal surface (Figure 4), and that the crown itself is not a free parallelepiped, but, together with its root, is part of a system containing supporting structures (i.e., the periodontal ligament and underlying bone).

A power analysis to evaluate the sample size was performed by using the G*Power software. The hypothesis that net force experienced by active aligners 1 and 2 is equal was analysed with a confidence level of 0.95, and the effect size of the sample was found to be d = 0.82 corresponding to a power of the test equal to *p* = 0.99.

A post hoc power analysis was performed on the comparison between aligners 2 and 3 as well; an effect size of d = 2.248 was found, corresponding to a power *p* = 0.997 for the hypothesis that the net force between the two aligners is different. If a priori analysis on the sample size is performed, a minimum sample dimension equal to 9 is found. These considerations justify, from a statistical point of view, the proper choice of the sample size. 

The difference in the response among the three passive aligners was tested from a statistical point of view. Figure 3 and Table 1 show some differences in several regions of the specimen. In fact, while no pressure is observed in areas B, C and D for all three of the specimens, differences are observed for that which concerns areas A and E. To test if the difference of observed pressure in area A in the P1 and P2 sample, a t-student analysis was performed on the set of ten measurements obtained for each sample. The t-value was found to be 4.16, so the recorded pressures can be considered different with a significance level *p* < 0.01. Analogously, if area E is compared for passive aligners P1 and P3, a t student value t = 6.65 is found, showing that recorded pressures in those areas can be considered different at a significance level *p* < 0.01.

Results about mean pressure recorded in the five regions of interest for the three active aligners are summarised in Table 2. The level of significance for all of the regions was tested for A1, A2 and A3, considering the ten measurements performed on the three specimens. For all of the three specimens, the t student test demonstrated a significant difference (*p* < 0.05) among the values observed in the three specimens, with the only exception of region C. As a further analysis, the test was repeated, with reference to region A and C, to calibrated values, as obtained by subtracting the passive aligner values from the active aligners. Additionally, in this case a significant difference (*p* < 0.05) was observed. 

Finally, a statistical comparison was performed on the calculated net force. Ten values of net force were compared for each active aligner. When the net force was compared for aligners A1 and A2, a t-student value t = 1.85 was found with a significance level *P* = 0.08. This shows that, even if a different distribution of the forces is present, it corresponds to the same level of net force. A different result is found if the third aligner is compared; in this case, a statistically significant difference (*p* < 0.01) can be observed.

## 4. Discussion

Although the literature presents several articles on in-vitro and in-vivo studies of the forces exerted by orthodontic appliances, it is difficult to compare our analysis with any of these, due to the innovative methodology we employed. Furthermore, the majority of studies considered the force applied, but not the pressure exerted. For example, Elkholy [14,15] studied the forces applied during palatal or labial movement of a central incisor (upper jaw model Frasaco, Tettnang, Germany) using a force/moment sensor (Nano 17, ATI Industrial Automation, Apex, NC, USA). The aligners studied by Elkholy were made of Duran^®^ of thickness 0.50, 0.625 and 0.75 mm. These yielded different force values, especially as regards the measurements of movements from the palatal side. Li [16] conducted a study using aligner materials and sensors different to ours, and even if we compare their results with those reported for the same thickness of the same aligner material at the same time-point by Elkholy, there are no similarities; indeed, the numerical variation is as high as 5 units. Although the sensors they used were different, this cannot entirely explain the discrepancies between their respective results.

Another study that we could consider is that by Hahn. They focused on central incisor torque, but reported values on three spatial planes, thereby allowing conversion of angular parameters into millimetres. As their torque measurements were expressed as palatal, vestibular and vertical intrusion forces, it is theoretically possible to compare the values they reported for vestibularisation and palatalisation with those reported by the abovementioned authors. However, even though the same sensor and unit of measurement were used, Hahn took measurements at time 0, while the others reported measurements taken after 30 min, 1 h or several days from aligner fitting, with obvious ramifications for the measurement of forces. Furthermore, the orthodontic movement considered by Hahn was 0.15 mm, while in the other studies it was 0.25 mm.

The study which is most similar to ours was reported in an article by Barbagallo [17] in 2007. They calculated the force exerted by an aligner during 0.5-mm vestibulo-lingual movement of a first premolar. They also used a pressure-sensitive film, Fuji Pressurex, but not the scanner we used to reveal the pressure values exerted on the film itself. Instead, they relied on a light microscope (SZ-FO Dissecting Microscope, Olympus, Japan) connected to a digital camera (Color View Soft Imaging System, Olympus, Japan), and the resulting images were then analysed by means of a spectrophotometer. Moreover, their study was conducted in-vivo, and they reported results as forces rather than pressure. Barbagallo et al. reported a mean force exerted by the aligner on the premolar was 28.09 ± 5.64 N at 15 s from its application. However, the authors subtracted from this raw figure a series of variables associated with the sensor, the anatomy of the analysed tooth and its anatomical location (although the details of this process were not fully specified).

Pressure films are extremely sensitive. Using them directly inside the oral cavity (without counting the thickness of the film itself) would have led to the creation of strengths that were not really present. Our intent is to describe the areas of applications and maximum pressures. Therefore, we know the perfect areas and we know that the pressures in the oral cavity, thanks to the periodontal ligament, will certainly be lower than those obtained. Hence, the historical concept of low and continuous forces in orthodontics. 

Moreover, it should be considered that, to avoid performing approximations of purely mathematical value and to focus instead on the raw measured data, our final values are reported in relation to the coronal area of the tooth analysed, which we treated as a parallelepiped (Figure 5). The net force on the tooth crown we calculated does therefore not wholly correspond to the force exerted by the aligner on a real tooth, which is supported by the surrounding soft and hard tissues. Indeed, the ultimate aim of our analysis was not so much to calculate the final force, but rather to determine whether “identical” aligners exerted the same force on the same tooth while performing the same movement. It should be observed that, when comparing aligners A1 and A2 a different distribution of the solicitations can be observed when the five regions of interest are considered. However, if the net force is analysed, the two samples behave in a similar way. Different observations should be made for sample A3, wherein a statistically significant difference is observed. This can be attributed to variations in the 3D printing and manufacturing of the aligners. Despite this, it should be noted that difference is nonetheless acceptable according to the orthodontics practice. 

This study, having been performed in vitro and not in vivo, has limitations due to the difficulty of reproducing the environment present in the oral cavity. This has always been dictated by the fact that the sensor is extremely sensitive and therefore its direct application in the oral cavity would have considerably reduced the level of precision in the measurements.

## 5. Conclusions

The areas of pressure exerted by the three aligners analysed were similar, but not perfectly superimposable and therefore not entirely reproducible, likely due to manufacturing and methodological variables. That being said, the pressure exerted on the tooth to be moved was almost identical across all tests conducted. On the whole, the pressure, and hence the forces exerted during orthodontic aligner treatment, appear to be controlled and reproducible, and have a mean value of roughly 15 MPa. 

## Figures and Tables

**Figure 1 nanomaterials-09-01010-f001:**
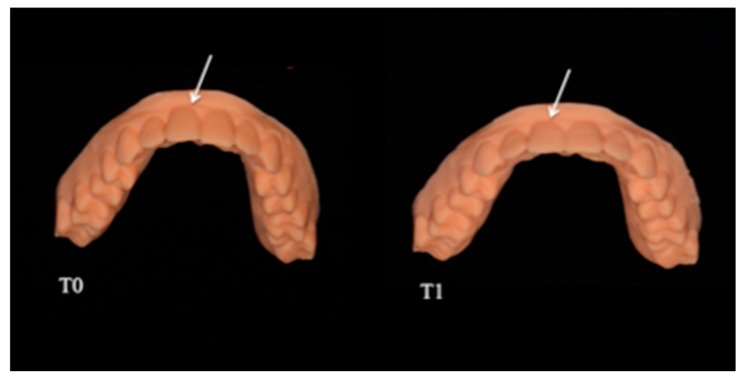
T0 and T1 models. The white arrows indicate the analysed element, before and after moving with aligners.

**Figure 2 nanomaterials-09-01010-f002:**
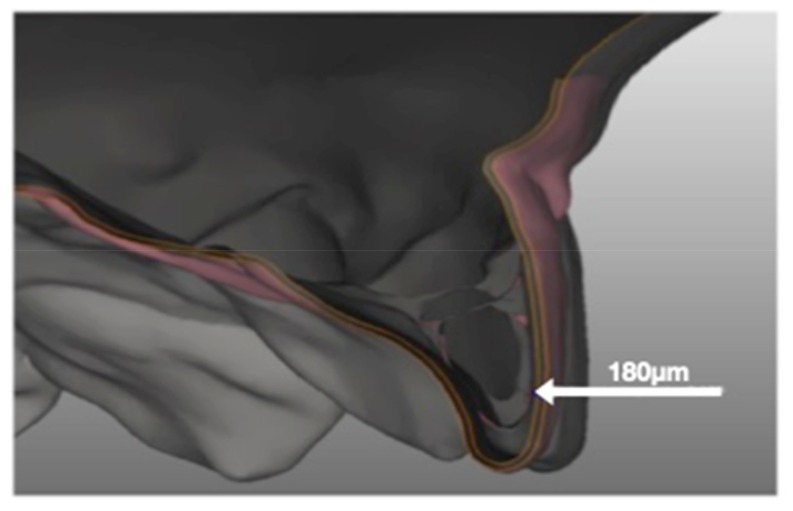
Sinergia software images generated with a view to creating the 180-micron pressure-film housing. The bright lines indicate exactly the thickness of 180 µm that has been digitally eliminated from the 3D model.

**Figure 3 nanomaterials-09-01010-f003:**
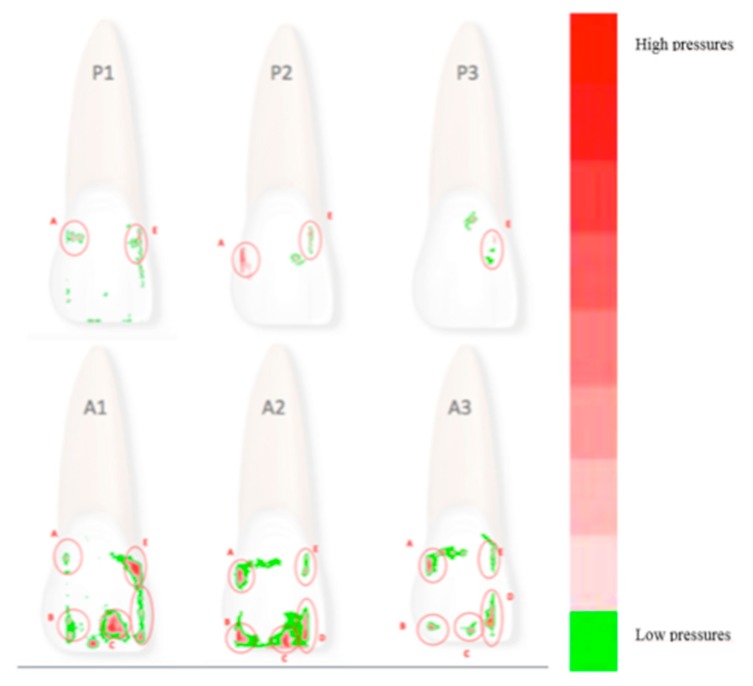
P1, P2 and P3: areas of pressure revealed by Super Low Pressure (SLP) film with passive aligners. A1-A2-A3: areas of pressure revealed by Low Pressure (LP) film with active aligners. The different colours correspond to different pressure values. The green scale indicates minimum pressure values while the red scale indicates maximum pressure values. The active values were later adjusted by subtracting the passive values for calibration purposes.

**Figure 4 nanomaterials-09-01010-f004:**
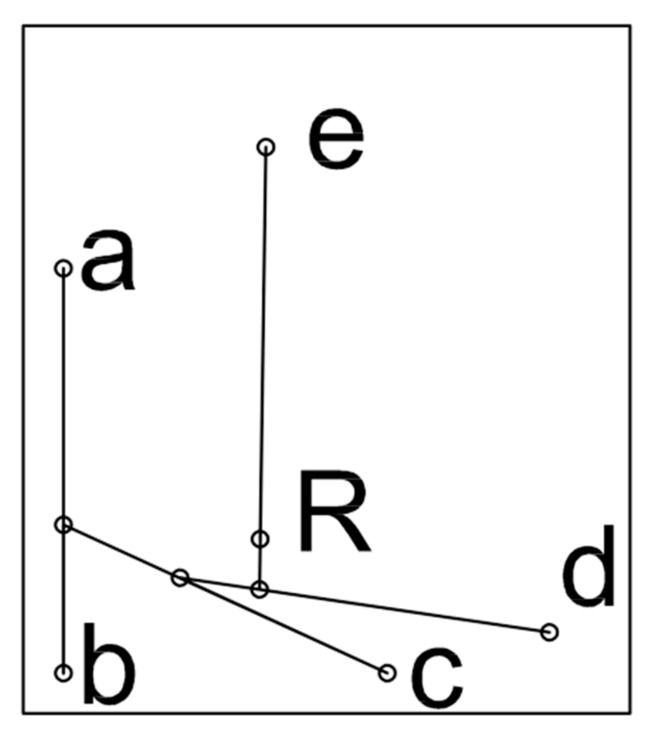
Calculation of the resulting force exerted by Aligner 2 on the vestibular surface of the tooth. The letters a, b, c, d, e, respectively indicate the points of application of the forces exerted by the pressure areas A, B, C, D, E, while the letter R indicates the point of application of the resulting total force exerted by the aligner on the tooth.

**Figure 5 nanomaterials-09-01010-f005:**
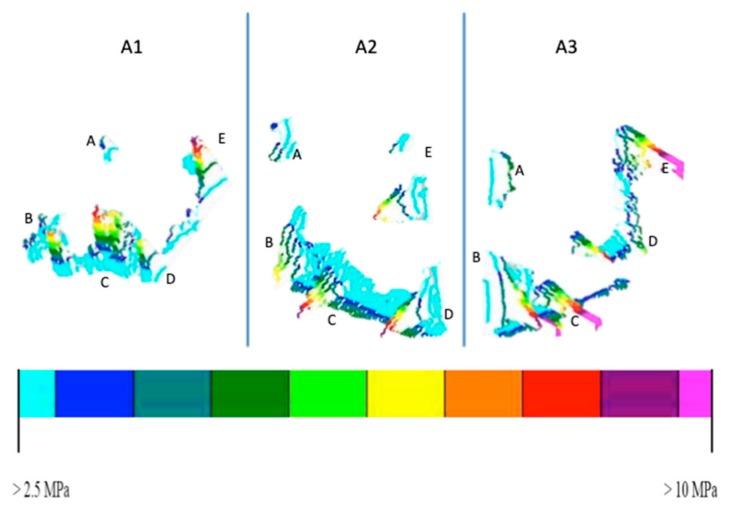
The 3D images of the 3 aligners, analysed as if they exerted pressure on the surface of parallelepiped. The colours shown in the figure indicate the various pressure variations. Green indicates lower pressures while yellow and red indicate higher pressures.

**Table 1 nanomaterials-09-01010-t001:** Mean and maximum pressure obtained for the three passive aligners (P1, P2 and P3) during the calibration phase.

	P1					P2					P3				
Pressure Areas	A	B	C	D	E	A	B	C	D	E	A	B	C	D	E
Mean pressure (MPa)	0.64	0.00	0.00	0.00	0.66	0.78	0.00	0.00	0.00	0.00	0.00	0.00	0.00	0.00	0.51
Maximum pressure (MPa)	1.29	0.00	0.00	0.00	1.34	1.46	0.00	0.00	0.00	0.06	0.00	0.00	0.00	0.00	1.01

**Table 2 nanomaterials-09-01010-t002:** Pressure values obtained for the three active aligners (A1, A2 and A3), raw data.

	A1					A2					A3				
Pressure Areas	A	B	C	D	E	A	B	C	D	E	A	B	C	D	E
Mean pressure exerted (raw data) (MPa)	3.43	2.65	3.43	3.26	2.96	1.56	3.05	3.91	3.48	2.63	2.06	5.58	4.05	2.43	4.53
Maximum pressure exerted (raw data) (MPa)	8.03	5.03	6.66	8.06	5.54	3.36	8.13	10.82	10.11	7.05	4.41	8.86	7.89	5.83	8.90

**Table 3 nanomaterials-09-01010-t003:** Mean values calibrated of the 3 active aligners. Active areas did not always have corresponding passive areas, which is why only some active pressure values are calibrated.

	A1					A2					A3				
Pressure Areas	A	B	C	D	E	A	B	C	D	E	A	B	C	D	E
Mean values calibrated (MPa)	2.79	2.65	3.43	3.26	2.30	0.78	3.05	3.91	3.48	2.63	2.06	5.58	4.05	2.43	4.02

**Table 4 nanomaterials-09-01010-t004:** Values for resulting force and their points of application on the three active aligners studied.

	Point of Application of Net Force	Modulus of Net Force
Aligner 1	(3; 3.11)	13.96 N
Aligner 2	(2.93; 2.15)	13.87 N
Aligner 3	(2.87; 2.44)	17.49 N

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
