# Peer review of "Experimental Study of the Pressures and Points of Application of the Forces Exerted between Aligner and Tooth"

_nanomaterials, 2019, doi:10.3390/nano9071010_

Reviewer 1 Report

Remark to the authors

This revised manuscript describes the experiment to investigate the pressure and points of forces mapped between aligners and tooth. The author utilized resin printed models at 2 timepoints from a patient scanned digital models. Two sets of 3 identical aligners were fabricated with thermoplastic polyurethane material and the pressure sensitive film were adapted into the aligner before the insertion of the printed model. The manuscript provides interesting information. The below questions has not been answered. 

1. No power analysis due to no statistical analysis. The study is a descriptive studies. I wonder if the author would attempt to analyze the result with any statistical analysis.

2. How accurate did the author apply the film into the inside of aligner? Since it is pressure sensitive, would it create the error of application? The author did not explain or discuss any prevention from the error of finger pressure or errors from other causes beside the pressure from the teeth. 

3. There is no discussion on limitation of the study.  Though the authors stated in the response regarding the effect of periodontal ligament but did not state anything in the discussion part. 

Overall this study is an interesting area of investigation however, certain information needs to be added for a better comprehensive elucidation and conclusion. 

Thank you very much for the invitation to review this manuscript.

Author Response

The application of the film, due to its high sensitivity and precision, was performed by means of a pin-type pliers. Furthermore, the sensor was cut with longer ends to avoid direct contact of the analyzed part.

Pressure films are extremely sensitive. Using them directly inside the oral cavity (without counting the thickness of the film itself) would have led to the creation of strengths that were not really present. Our intent is to describe the areas of applications and maximum pressures. Therefore, we know the perfect areas and we know that the pressures in the oral cavity, thanks to the periodontal ligament, will certainly be lower than those obtained. Hence, the historical concept of low and continuous forces in orthodontics. 

This study having been performed in vitro and not in vivo, presents limits due to the difficulty of reproducing the environment present in the oral cavity. This has always been dictated by the fact that the sensor is extremely sensitive and therefore its direct application in the oral cavity would have considerably reduced the level of precision of the measurements

Reviewer 2 Report

For me it is still not justified to have a row in table 3 with a heading "mean values calibrated" when only 4 out 15 values are actually calibrated. That means Table 3 mostly repeats the "raw data" of Table 2. Table 2 and 3 should be combined. Furthermore it looks to me that right the value A1E should be 2.30 MPa (instead of 2.90 MPa).

There are still a number of typos and editing mistakes throughout the text. There is no consistent style for the references.

Author Response

We performed the calibration based on the average values obtained. All values have been calibrated, but often the pressure value recorded for passive aligners was extremely small and therefore not relevant (average calibration value<0.00). This is the reason why only 4 values are calibrated.

Round  2

Reviewer 1 Report

This revised manuscript describes the experiment to investigate the pressure and points of forces mapped between aligners and tooth. The author utilized resin printed models at 2 timepoints from a patient scanned digital models. Two sets of 3 identical aligners were fabricated with thermoplastic polyurethane material and the pressure sensitive film were adapted into the aligner before the insertion of the printed model. The manuscript provides interesting information. The below questions has not been answered. 

1. No power analysis to verify the number of samples. 

Thank you very much for the invitation to review this manuscript.

Author Response

A power analysis to evaluate the sample size was performed by using the G*Power software. The hypothesis that net force experienced by active aligners 1 and 2 is equal was analyzed with a confidence level of 0.95, the effect size of the sample was found to be d=0.82 corresponding to a power of the test equal to p=0.99.

A post hoc power analysis was performed on the comparison between aligners 2 and 3 as well; it was found an effect size d=2.248 corresponding to a power p=0.997 for the hypothesis that the net force between the two aligners is different. If a priori analysis on the sample size is performed a minimum sample dimension equal to 9 is found. These considerations justify, from a statistical point of view, the proper choice of the sample size.

Nanomaterials EISSN 2079-4991 Published by MDPI AG, Basel, Switzerland RSS E-Mail Table of Contents Alert
Back to Top